# Electrosynthesis of Biobased Chemicals Using Carbohydrates as a Feedstock

**DOI:** 10.3390/molecules25163712

**Published:** 2020-08-14

**Authors:** Vincent Vedovato, Karolien Vanbroekhoven, Deepak Pant, Joost Helsen

**Affiliations:** Separation & Conversion Technology, Flemish Institute for Technological Research, Boeretang 200, 2400 Mol, Belgium; vincent.vedovato@vito.be (V.V.); karolien.vanbroekhoven@vito.be (K.V.)

**Keywords:** electrosynthesis, biomass, carbohydrate, saccharides, electro-oxidation, electroreduction

## Abstract

The current climate awareness coupled with increased focus on renewable energy and biobased chemicals have led to an increased demand for such biomass derived products. Electrosynthesis is a relatively new approach that allows a shift from conventional fossil-based chemistry towards a new model of a real sustainable chemistry that allows to use the excess renewable electricity to convert biobased feedstock into base and commodity chemicals. The electrosynthesis approach is expected to increase the production efficiency and minimize negative health for the workers and environmental impact all along the value chain. In this review, we discuss the various electrosynthesis approaches that have been applied on carbohydrate biomass specifically to produce valuable chemicals. The studies on the electro-oxidation of saccharides have mostly targeted the oxidation of the primary alcohol groups to form the corresponding uronic acids, with Au or TEMPO as the active electrocatalysts. The investigations on electroreduction of saccharides focused on the reduction of the aldehyde groups to the corresponding alcohols, using a variety of metal electrodes. Both oxidation and reduction pathways are elaborated here with most recent examples. Further recommendations have been made about the research needs, choice of electrocatalyst and electrolyte as well as upscaling the technology.

## 1. Introduction

The increasing pressure on non-renewable fossil resources caused by the growing population has made increasingly urgent the need for the development of sustainable and alternative technologies to produce energy, food, and materials. In recent years, a significant number of environmentally friendly processes based on the utilization of renewable raw materials have been described, in the hope of replacing the petro-based industry [1]. This is especially true of Europe where bio-based industries are central to build a European circular economy [2].

Biomass is considered as an inexpensive feedstock and is the most abundant renewable resource on the planet. It is a mixture composed of organic and inorganic materials. One of the major components of the organic biomass materials is lignocellulose, itself being constituted primarily of cellulose (42–49%); hemicellulose (16–23%), and lignin (21–39%) [3]. Cellulose is thus considered to be the most important macromolecule on Earth. It is composed of long polymeric chains of at least 500 glucose molecules. This abundance explains the renewed interest towards saccharides and polysaccharides as an alternative to fossil resources in chemical, pharmaceutical, material, and energy industries. Even though carbohydrate chemistry has been a subject of study since the 19th century following Fisher’s pioneering work [4], it has arguably received limited attention compared to other biomolecules such as proteins or amino-acids. While these compounds have found a plethora of applications in various industries, their production and derivatization are predominantly via chemical synthesis. Enzymatic processes have been studied as a greener alternative but remain yet underexplored [5,6]. Therefore, there is a great need for novel greener processes for the production of carbohydrate-based materials.

The field of electrochemistry has recently gained renewed traction as a clean and carbon-neutral way to promote chemical transformations. Electrochemistry can be employed in a large number of areas, from the synthesis of materials or chemicals to the development of sensors to detect and analyse specific compounds, as well as to generate power (cogeneration). In chemical synthesis, electrochemistry allows to avoid the use of chemicals to deliver the reductive or oxidative power and provides more safe processing conditions due to the decoupling of the chemical reaction in half reactions [7]. Although electrosynthesis satisfies most of the postulates of green chemistry, direct applications of such processes in industries remain rather scarce. However, recent advancements in the development of electrode materials and membrane technologies have improved the performances of electrochemical processes by reducing the energy consumption, improving the rates of reactions and selectivity and increasing the current density [8,9]. These novel technologies and the ability to use renewable electricity from wind or solar energy have made electrochemistry an intensively researched approach in recent years which has a potential to become less expensive and therefore a more economically and ecologically attractive alternative to chemical processes.

This review addresses the current knowledge on the electrochemical conversion of biobased chemicals, particularly saccharides, to commodity chemicals.

## 2. Electrochemical Oxidation of Biomass

The oxidation of carbohydrates has been a very active field of research for the last fifty years and is still intensively studied. This is mainly true for oxidation of carbohydrates to their uronic acid analogues. Chemical direct oxidation processes require the use of stochiometric amount of strong and toxic oxidizing agents, such as nitric acid and nitrogen dioxide. Due to the risks and cost of such protocols, catalytic processes were developed, employing atmospheric oxygen and noble metals. Although the catalytic systems were more selective and efficient, more sustainable systems were greatly sought. Therefore, many reports have demonstrated the capacity for electrochemical processes to transform saccharides—through direct or indirect oxidation—to their corresponding uronic acids, but also to other important derivatives (Scheme 1). Reducing and non-reducing saccharides have been studied. These two types of saccharides differ in the presence or the absence of a free aldehyde/hemiacetal functional group.

### 2.1. Poly/Oligosaccharides

The organic stable nitroxyl radical 2,2,6,6-tetramethylpiperidin-1-oxyl (TEMPO, **1**) is used commonly in organic synthesis, mostly for its ability to oxidize primary alcohols with a high level of selectivity. Due to the high turnover frequency of TEMPO and its derivatives, as well as their high stability, such catalyst also found use in electrochemical processes, as homogeneous mediators for electrochemical oxidation of hydroxyl groups (Scheme 2).

Polymeric materials with interesting physicochemical properties which, after their period of utilization, degrade to green components are highly sought. Natural polymers have been used as a starting platform to study the feasibility of such concepts by modifying their primary structure.

Being naturally present in most plants for energy storage, starch is produced on a multi-million tons scale per year and has found a great number of applications in the food industry (starch is naturally present in many basic cooking ingredients, but is also used as thickening agent) and mainly in the papermaking process as an adhesive.

The chemical oxidation of carbohydrates using nitrogen oxides is known since the mid-1900′s, however, such transformation always requires the use of external, toxic co-oxidants, in order to regenerate the nitrosonium ion **2**, from the hydroxylamine **3**. In order to develop a more sustainable and user friendly process, Schafer and co. reported a selective TEMPO mediated electrochemical oxidation of potato starch **4** to its corresponding α-d-glucuronan **5** (Scheme 3) [10]. This electro-oxidation of potato starch was highly selective toward the primary hydroxyl group in the 6-position. The corresponding polyuronic acid was isolated in 63% yield as a crude product with a 93% conversion of primary alcohol groups.

The electrochemical transformation delivered a slightly lower yield of crude product compared to the chemical reaction (>95% selectivity and 98% crude product yield), however, it avoided the need to employ sodium hypochlorite, previously used in stoichiometric amount to regenerate the active TEMPO species.

The development of novel electrochemical processes also allows for new applications to be discovered. Dang et al. reported a one pot electrochemical oxidizing coupling of corn starch and natural gelatin [11]. This oxidized corn starch-gelatin material was then dehydrated and granulated with the spray-dying process. After full characterization, the authors reported that such material could be used for drug delivery applications.

Cellulose is the most abundant biopolymer on the planet and due to its high level of functionalization, it is a compound of great interest for a vast number of applications. Electrochemical transformations of cellulosic materials have been studied mainly toward the development of biobased fuel cells [12] in order to provide sustainable electricity. However, in such processes, the outcome of the electrochemical oxidation reaction is of minor importance and the selectivity of the reaction is in most cases not studied, therefore, such processes are beyond the scope of this review. Recently, Yang et al. documented a controlled electrochemical transformation of cellulose oligosaccharides **6** into glucose (**7**) [13]. A hydrothermal pre-treatment of cotton cellulose was first necessary to isolate the desired starting material oligosaccharides. The optimal reaction condition for the electrochemical transformation of the obtained oligosaccharides could then be determined. The process which showed the most reactivity and selectivity toward the formation of glucose was a direct electro-oxidation on a MnO_2_/graphite/PTFE anode, in acid media (pH = 3.0). Under this optimized reaction conditions, glucose was formed in 72% yield with 100% selectivity (Scheme 4).

The electrochemical oxidation of cellulose has also been reported using a gold electrode, in alkaline media (1.3 M NaOH) [12,14]. Although the oxidized products were not clearly identified, it is worth noting that the authors proposed a detailed mechanism to explain the interaction between cellulose and the various gold species formed at the surface of the electrode. Indeed, each step of adsorption, oxidation, desorption of the cellulose could be identified and associated to specific gold species. The products generated from this reaction are presented as one water soluble material in which some hydroxyl groups were oxidized to carboxylic acids, and one water insoluble hybrid material made of cellulose and gold nanoparticles.

A similar study was then performed on the electrochemical oxidation of hemicelluloses materials at a gold electrode [14]. The authors analysed the responses of various hemicelluloses (xylan, arabinoxylan, glucomannan, xyloglucan, and glucuronoxylan) in alkaline solution (1.3 M NaOH solution). Although xylan, xyloglucan and glucuronoxylan were found to be electrochemically active, only xylan was studied in more details and was submitted to a long-term electrolysis. The analysis of the products form xylan electrolysis showed a water-soluble material in which OH-groups were oxidized to carboxylic groups and a xylan-based water insoluble material containing gold nanoparticles.

Raffinose (**8**), a trisaccharide mostly found in cotton seed, is constituted of galactose, glucose, and fructose units. The electrochemical oxidation of raffinose was reported using TEMPO as a mediator, to selectively oxidize the primary alcohol moieties, while leaving untouched all secondary alcohol groups present on the molecule. The electrochemically oxidized raffinose was then esterified to trimethyl d-raffinose trisuronate (**9**) and isolated in 63% yield (Scheme 5) [15].

### 2.2. Disaccharides

Sucrose is a non-reducing disaccharide constituted of d-glucose and d-fructose units which are glycosidically bonded through their anomeric carbon atoms. Sucrose (or saccharose) is broadly present in plants and is the main disaccharide feedstock across the globe. It is an important carbohydrate reserve compound and one of the main energy sources required in the human’s diet. It is produced from sugar cane and sugar beet (130 × 10^6^ t/year) [16] and is present in honey, maple syrup, fruits, and vegetables. Sucrose itself is widely used in the food industry for its sweetening ability. Sucrose can be functionalized at various positions and therefore can be converted to various interesting additives. Monocarboxylic acids of sucrose are valuable intermediates in the synthesis of monoesters and monoamides of sucrose, which are used as for their tensioactive capacities. However, examples of chemical and electrochemical transformations of sucrose are scarce due to the poor selectivity of the reactions.

One of the most noticeable examples of electrosynthesis employing disaccharides as starting material was reported by Lamy and co-authors in 1993 [17]. The authors documented a thorough study of the electrochemical behaviour of sucrose in various reaction conditions, as well as the capacity of a range of electrodes to perform the desired oxidation reaction. Due to the known capacity of saccharose to hydrolyse under acidic condition an alkaline media was therefore used to conduct the experiments. A vast number of electrode materials were tested and evaluated by using voltammetry analysis. Ir, Fe, and Co showed no activity toward the oxidation of sucrose; Cu, Ag, Ni, Rh, and Pd displayed some activity but too weak to be explored further. Lastly, Au and Pt showed good reactivity for the oxidation of sucrose. Gold produced a much higher current density than Pt, but required a much higher working potential for the oxidation to occur. The authors then turned their attention to parameters such as temperature and initial sucrose concentration. They used a potential step program, designed to clean the electrode of species which could remain adsorbed to the electrode surface. With the optimal reaction conditions at hand, a long term electrolysis was performed on a large Au plate electrode with sucrose (10 mM) in NaOH solution (0.1 M) for 12.5 h, aiming for the specific oxidation of primary alcohol groups, without cleaving the C-O-C bond. After 12.5 h of electrolysis, 67% of sucrose was converted to a range of products (Table 1).

In a follow up work, the same group was able to scale up their reaction by adapting it to a flow cell reactor [18]. The working electrode used in this novel system was a lead-modified platinum electrode. A similar potential step program as previously reported was employed in order to diminish to a maximum the electrode poisoning during the reaction, and therefore maintaining the high activity and selectivity of the electrocatalyst. After 8.5 h, the electrolysis was stopped and the products of the reaction analysed by different characterisation methods. The overall sucrose conversion was estimated to 60%, with a selectivity of 80% towards 6-monoacid of sucrose (**10**) and 10% toward the 1′-monoacid of sucrose (**11**). In order to confirm the structure of the reaction products, compounds **10** and **11** were hydrolysed and the resulting monosaccharide derivatives were analysed (Scheme 6).

Hence, the authors were able to assign with certainty the structure and the selectivity of the electrochemical oxidative process.

A mediated electro-oxidation of sucrose was also demonstrated by employing TEMPO as the homogenous mediator [10]. Within the structure of sucrose, the three primary alcohols groups are the most likely to be oxidised by TEMPO, due to its high selectivity toward primary alcohol oxidation. When the electrolysis was stopped after the consumption of 6970 C (4.8 F/mol), 83% sucrose was converted and the three possible sucrose monocarboxylic acid species (such as **10** and **11**) could be observed in various amounts (low to moderate yields). However, when the electrolysis was stopped after 28,450 C (19.7 F/mol) were consumed, only the tricarboxylic acid species could (**16**) be observed and isolated in 39% yield (Scheme 7).

Other disaccharides were also submitted to electrochemical transformations. Trehalose is a non-reducing sugar constituted of two d-glucose units, linked by a α(1→1) glycosidic bond. Both direct and indirect electrochemical oxidation of trehalose were studied. Schnatbaum et al. applied a similar reaction procedure than for the electrochemical oxidation of sucrose, and were able to demonstrate the ability of TEMPO to act as a selective mediator for the oxidation of α,α-trehalose (**17**) toward α,α-trehalose dicarboxylic acid (**18**) [10]. The desired product was isolated in 61% yield after consumption of 2010 C (8.8 F/mol). In a later study, Kokoh and co-workers were able to further improve the TEMPO-mediated electrochemical oxidation of trehalose by fine tuning the electrolysis potential They were also able to improve the analytical process by transforming **18** into its trimethylsilylated equivalent for a better detection by GC-MS [19]. Using this processes, the authors were able to report a full conversion of trehalose (**17**) to the dicarboxylic product (**18**).

Parpot et al. studied a direct electrochemical oxidation of trehalose using a gold electrode or a gold-nickel (70/30) alloy, in alkaline media [20]. It was postulated that combining gold and nickel in an alloy would produce a great selectivity towards the oxidation of primary alcohols (due to gold), and high current density because of the nickel atoms. Indeed, saccharide oxidation at nickel electrode is reported to occur at a high anodic potential, in NiOOH region, but generates low molecular weight carboxylic acid species. The authors first noticed that trehalose was not showing significant oxidation signals in carbonate buffer. They attributed this lack of reactivity to the absence of aldehyde/hemiacetal group, which are found in reducing sugars. Considering this aspect, the authors report changing the electrolyte from carbonate to NaOH (0.1 M) was significantly beneficial to the reactivity of trehalose with Au and Au/Ni electrodes. In order to maximise the efficiency of the electrodes, the authors used a pulse potential program for long-term electrolysis. The aim of this potential program was to perform the oxidation at potential 0.25 V vs. RHE for 60 s, followed by a cleaning pulse at 1.7 V vs. RHE for 1 s. This pulse was used to remove any species which might have remained adsorbed on the surface of the electrode after the oxidation reaction was done. The end products of the reactions were characterised by GC-MS and LC-MS. Independently of the electrode used, the main product was the trehalose monocarboxylic acid (**19**). Alongside this product, significant amount of trehalose dicarboxylic acid (**17**) and glucose were also detected, as well as other acids of low molecular masses (Table 2).

This study showed no significant difference between a pure gold electrode and a gold/nickel alloy electrode toward the selectivity or the efficiency of the reaction (Table 2) [21,22].

Reducing sugars are carbohydrates with an unprotected aldehyde or hemiacetal group. The reactivity order of the various groups present in reducing sugars toward an oxidative process are as follow: equatorial secondary hydroxy < axial secondary hydroxy < primary hydroxy < hemiacetal hydroxy. Therefore, the unprotected hemiacetal group present in disaccharides such as cellobiose, d-maltose, and d-lactose is the most likely to react first, which could bring further selectivity issues than with non-reducing sugars.

Cellobiose (**19**) is constituted by two glucose units, linked by a β(1→4) glycosidic bond. It can be obtained from materials such as cotton, jute, or paper after hydrolysis of the cellulose. d-maltose (**23**) is a stereoisomer of cellobiose, formed of two glucose units which are is this case linked by an α(1→4) bond. Due to their similar structure, cellobiose and maltose were often studied in parallel. Both disaccharides were submitted to similar electromediated oxidation processes by Liaigre et al. [23]. Both **19** and **23** were oxidised electrochemically using TEMPO as a homogeneous mediator, and in alkaline media (1.0 M carbonated buffer). However, the selectivity of these transformations toward the corresponding triacid products (**20** and **24** respectively) were very low when the reactions were performed at room temperature, and lead to compounds of low molecular masses such as tartaric acid and oxalic acid. Cooling the reaction mixture to 2 °C improved significantly the selectivity of the reaction and the desired tricarboxylic acid products **20** and **24** were reported in moderate to good yields (Scheme 8).

In a subsequent study Schämann et al. reported the scaling up of the electrochemical oxidation process of d-maltose (up to 67.5 mmol), and the further transformation of the tricarboxylic acid in the corresponding trimethyl ester in 50% yield [15,24]. In a comparative study between reductive and non-reductive sugars, Parpot et al. were able to demonstrate that the selectivity of TEMPO for the electro-oxidation of **23** was lower than the one observed for trehalose [19]. They attributed this to the presence of the free aldehyde group. During their investigation the authors were able to detect by GC-MS the low molecular weight compounds, but also mono-, di-, and tricarboxylic acid products, in significant amounts.

d-Lactose (**25**) is also a reducing sugar, it is formed of a galactose and a d-glucose subunit, linked by a β(1→4) glycosidic bond. Lactose is naturally present in milk, in 2 to 8% by weight and is thus available in large quantities (about 300,000 t/year). Its TEMPO-electromediated oxidation was also studied by Shafer and co-workers [15]. d-Lactose was electro-oxidised in alkaline media (NaHCO_3_/Na_2_CO_3_) at constant potential (E = 0.77 V vs. RHE). Under these reaction conditions, d-lactose was consumed quantitatively, however, the selectivity of the reaction was poor. Indeed, five different products could be identified by ESI-MS but were not isolated (Scheme 9). The expected mono, di, and tricarboxylic acid products **26-28** were present, as well as the diketo product **29**. The monosaccharide galactaric acid (**30**) was also identified, and arises from the cleavage of the β(1→4) glycosidic bond. This phenomenon was also observed during the TEMPO electromediated oxidation of d-cellobiose (**19**).

The electrochemical behaviour of disaccharides has also been studied in direct oxidation processes on rare metal electrodes. As previously mentioned, a direct electrochemical transformation gives more insights on the reaction itself, and in theory, allows a better control on the reaction by being able to tune the potential at which the transformation is occurring. Whilst a wide range of metal electrodes were tested for the electrochemical oxidation of disaccharides (Au, Pt, Ni, Cu, Co, Ru, Cd, Ir), most reports only documented voltammetric studies, evaluating the electrical response of disaccharides on the metal electrode. These types of studies were looking for application in sensors, and did not report the products of the reaction. Only few reports are giving information on the reaction products formed after a long term electrolysis.

Druliolle et al. studied the oxidation of d-Lactose (**25**) on platinum and on modified platinum electrodes in alkaline medium toward the formation of lactobionic acid (**26**) [25,26]. The authors first studied the electrocatalytic ability of a smooth platinum electrode to oxidise lactose in a sodium hydroxide solution (0.1 M), at room temperature using cyclic voltammetry (CV). However, from this preliminary tests, lactose did not demonstrate good electrochemical reactivity on platinum and no lactose was oxidised during a long term electrolysis. Then, they evaluated the effect of different underpotential deposited (upd) metal adatoms. Indeed, the ability of upd adatom to enhance the catalytic activity of noble metals for electro-transformation of organic compounds is well documented [27,28,29,30]. The authors studied the catalytic effect of Tl, Bi, and Pb adatoms on a platinum electrode in 10 mM of lactose and 0.1 M NaOH. The CV measurements showed some significant differences in terms of optimal oxidation potential and some drastic increase of current density.

Long term electrolysis of lactose was carried out with bismuth perchlorate salt in NaOH solution (0.1 M). A potential step program was used to maximise the rate of the oxidation reaction, while depositing the Bi metal adatoms onto the electrode, and minimising the poisoning of the electrode. After two hours of electrolysis, the recorded quantity of electricity passed was Q_exp_ = 20.7 C. After analysing the crude material of the reaction by HPLC, the authors reported that lactose was converted in 70% exclusively to lactobionic acid (**26**), and that no other products could be detected by their analytical method, therefore awarding a nearly 100% selectivity toward **26**, with a Faradaic yield of 75%. The lower Faradaic yield compared to the overall selectivity of the reaction can be explained by the slow decomposition of **26** under the reaction conditions.

Long term electrolysis of d-Lactose on lead adatoms modified platinum electrode in carbonated buffer was also studied [26]. A similar potential step program was used, with the adatom deposition plateau, the oxidative plateau, and the final step to remove the poisoning species from the surface of the electrode. After three hours of electrolysis, the recorded quantity of electricity passed was Q_exp_ = 15.1 C. After analysing the resulting mixture, the authors report that 78% of the starting lactose was converted to the desired lactobionic acid with nearly 100% selectivity. If the reaction was continued to reach four hours, 95% of lactose was converted, but the selectivity toward **26** decreased to 76%, which can be explained by the slow decomposition of lactobionic acid under the reaction conditions.

Druliolle et al. then turned their attention to gold electrode for the oxidation of d-Lactose (**25**) [31]. The ability of gold based electrocatalyst to oxidise selectively alcohol groups in alkaline medium was previously reported by Larew and co-workers [32]. Furthermore, gold is known to be less subject to poisoning than palladium, which explain the increase of attention which was directed toward this noble-metal for electrochemical transformations. To determine the optimal reaction conditions for the electro-oxidation of lactose on a gold electrode, the authors used voltammetric measurements. They report that a potential step program was used to maximise the oxidation of **25**, while minimising the poisoning of the gold electrode, under alkaline medium (carbonated buffer, 0.1 M). Then, the authors evaluated the impact of the starting concentration of d-Lactose, as well as the effect of small variation on the oxidative potential plateau. With the optimal reaction conditions ([lactose] = 10 mM) in hand, 95% of **25** was converted to lactobionic acid, with 98% selectivity after five hours of electrolysis.

By using two complementary IR analytical methods (SPAIRS and SNIFTIRS), the authors were able to identify the possible steps of the reaction, and therefore to propose a possible mechanism for the electro-oxidation of d-Lactose at a gold electrode (Scheme 10). During this investigation, the authors noticed that two different signals, depending on the oxidative potential applied. When the applied potential of the reaction is below 1.1 V vs. RHE, a characteristic signal of vibration mode was observed, which can be attributed to an asymmetric O-C-O bond stretching (Scheme 9, left hand side cycle). At higher potential, a signal characteristic of a symmetric O-C-O stretching (Scheme 9, right hand side cycle) bond could be identified. Gold being known for its oxygen adsorption properties, it was assumed that at low potential, a small quantity of hydroxyl species can be adsorbed on the gold electrode, and at higher potential, a much larger quantity of gold hydroxide species could be generated. Despite the differences observed by infra-red spectroscopy, only lactobionic acid was observed after the reaction, which indicate that at different potential, different mechanisms are taking places, yet delivering the same product. The authors postulated that once d-Lactose was adsorbed on the gold surface, either a hydroxide group could attack on the glycosidic carbon, therefore opening the ring (Scheme 9, left hand side cycle), or if the concentration of gold hydroxide species was sufficient, another Au-OH group could open the ring (Scheme 9, right hand side cycle). Both intermediate could then deliver the desired product **26**.

In a following study, Kokoh et al. documented the electrocatalytic oxidation of d-Lactose on nanoparticles of Au-colloids (5 nm), immobilised on a carbon felt electrode, in a flow reactor [33]. The optimal reaction conditions were first determined using CV, which demonstrated that the gold nanoparticles were able to generate a higher current density than when a plain gold electrode of three times the size was used. With the information gathered during the preliminary investigation, the authors determined that a two potential plateau program had to be employed to limit the poisoning of the electrode, as well as to maximise the oxidation reaction. The electro-oxidation was then carried out in carbonated buffer (0.1 M, pH = 10.2), and delivered lactobionic acid (**26**) in 91% yield.

Recently, Parpot et al. described the electrochemical oxidation of cellobiose (**19**) and d-maltose (**23**) on a gold electrode in alkaline media [20]. For each disaccharide, the author conducted preliminary investigations using CV. From these tests, it was possible to determine the optimal reaction conditions. As previously mentioned, gold electrodes are subject to poisoning. Therefore, in order to diminish this undesired phenomenon, a multi-potential plateau program was used for the long term electrolysis of both cellobiose and d-maltose.

With the optimal reaction conditions at hand, 95% of cellobiose (**19**) could be converted almost exclusively to its monocarboxylic acid equivalent, cellobionic acid (**27**), resulting from the oxidation of the aldehyde at carbon C1. The dicarboxylic acid product could also be detected in very small amount (3%) as well as trace amounts of the products resulting from C-C bond cleavage (Scheme 11A).

Then, d-maltose was submitted to its corresponding optimal reaction conditions for long term electrolysis. However, when d-maltose was submitted to electro-oxidation in NaOH solution (0.1 M), the reaction proved to be unselective, producing a variety of small carboxylic acids, such as tartaric acid, formic acid, etc., in non-negligible quantities. Furthermore only 50% of the initial d-maltose was converted after eight hours of electrolysis. A second electrolysis of d-maltose was then carried out in carbonated buffer (0.1 M, pH = 10), using the same potential plateau program. After twenty hours, 79% of the initial amount of d-maltose (**20**) was converted, with a selectivity up to 92% toward maltobionic acid (**28**), with only 3% of the dicarboxylic acid equivalent being detected, and no degradation products (Scheme 11B).

### 2.3. Monosaccharides

Significant interest has been drawn to the oxidation reaction of monosaccharides due to the variety of high-added-value chemicals which could potentially be obtained. Enzymatic and chemical processes, stoichiometric and catalytic, as well as homogeneous and heterogeneous catalytic systems have been reported for such transformations. However, these processes are beyond the scope of this work, and several reviews focusing on those aspects have been documented [34,35,36]. Concerning the electrochemical transformation of monosaccharides, the amount of reports is scarce. Most of the existing literature is focusing on the development of glucose biosensors to help for diabetic treatment, or as a reagent for fuel cells for clean energy production. However, in such studies, the structure of the products generated are generally not investigated. Amongst the electrosynthesis reports of the transformation of monosaccharides, d-glucose was the most widely investigated because it is one of the most abundant bio-based chemicals, and it can be converted to a variety of commodity chemicals [35,37]. In 1910, Lob described one of the first electrochemical oxidation of d-glucose, in sulfuric acid, with a lead anode and a platinum cathode [38]. The various products obtained were d-arabinonic acid and d-glucaric acid, as well as some fragmentation by products such as d-arabinose and formaldehyde. The slow degradation of d-glucose to lower aldoses during an electrolysis in non-aqueous media was investigated in more details by Hay et al. [39] although, the lack of detailed information such as yield, efficiency, and selectivity are limiting the comparative analysis.

Isbell and co-workers reported series of efficient indirect electrochemical oxidation of various monosaccharides into their corresponding aldonic acid salts [40,41,42,43]. Their process is based on the electro-oxidation of sodium or calcium bromide, generating free bromine at the anode, which reacts with the saccharide to give the corresponding aldonic acid. The calcium carbonate present in the reaction mixture quenches the acid, maintaining a constant pH (6.2) during the electrolysis (Table 3). The bromide is continuously regenerated, allowing the conversion of a large amount of sugars while using only a catalytic amount of bromide salt [40,41]. Most of the aldonic acid salts then crystallises and the acid form can then be recovered by reacidification and recrystallization [42]. The acid salt products could also be converted to the corresponding δ- or γ-lactones via an acid catalysed process.

When xylose was submitted to the electro-oxidation, the current density and the conversion of the starting material were very promising. However, calcium xylonate appeared to be very soluble, and therefore could not be separated by crystallisation in an efficient manner. The authors then applied similar reaction conditions for the electro-oxidation of xylose, but in the presence of strontium or magnesium bromide and carbonate. This allowed them to isolate the corresponding xylonate salt in high yields 49% and 80% respectively (Table 3, entry 5) [43].

In a follow up study, the same authors, and others groups, proposed a semi-industrial procedure for the production of calcium d-gluconate [41,44]. In this process, the calcium d-gluconate was recovered after the addition of calcium hydroxide, which generate a less soluble basic salt species. Despite continuing the electrolysis longer than theoretically needed to oxidise glucose to gluconic acid, a reducing compound remained in solution. After repeating Isbell’s procedure for an electrolysis, Cook et al. were able to isolate the calcium salt of 5-keto-gluconic acid (**36**) in approximately 5% yield [45]; therefore proving that gluconic acid was not the only product of the CaBr_2_ mediated electro-oxidation of glucose (Scheme 12).

Inspired by the examples of chemical oxidation of simple alcohols and carbohydrates by ruthenium complex catalysts [46]. Kokoh and co-workers decided to explore the possibility for a selective electro-oxidation of glucose using Ru-based complex catalysts [47]. Azopyridine (azpy) ruthenium complexes are known to maintain the ruthenium metal at a low redox potential state Ru(IV), which could be suitable for a selective electrochemical oxidation of carbohydrates [46,48,49,50,51]. The authors reported the synthesis of two isomers γ-RuCl_2_(azpy)_2_ and δ-RuCl_2_(azpy)_2_ and their immobilisation on carbon fibre materials prior to their electrochemical characterisation. Using CV measurements on crude RuCl_2_(azpy)_2_, the authors could determine that the redox couple observed in the positive scans was Ru(IV)/Ru(III). Voltammograms of the crude ruthenium complex recorded in the presence on d-glucose in alkaline medium (0.1 M NaHCO_3_/Na_2_CO_3_) showed no reduction peak, thus suggesting a fast reaction between the Ru(IV) and glucose. After having proved that the ruthenium complex was able to react with glucose, the authors set out to perform long term electrolysis. Using their optimal reaction conditions, d-glucose was electro-oxidised during 45 h at fixed potential in presence of the crude RuCl_2_(azpy)_2_ catalyst. Analysis of the crude reaction mixture showed a 22% conversion of glucose toward gluconic acid (**29**, 11% yield, 50% selectivity) and 2-keto-gluconic acid (**37**, 7% yield, 33% selectivity). Trace amounts of a tricarboxylic acid could also be detected by LC-MS analysis, as well as oxalic and tartaric acid. The presence of these low molecular weights acids suggested that the tricarboxylic acid was hydrolysed which resulted in the C2-C3 bond cleavage [52]. The authors postulated that glucose is getting oxidised to gluconic acid (**29**) via the intermediate δ-gluconolactone, but that the gluconic acid can itself be further oxidised and therefore deliver 2-keto-gluconic acid (Scheme 13).

Long term electrolyses were carried out using pure γ-RuCl_2_(azpy)_2_ and δ-RuCl_2_(azpy)_2_, using the same reaction conditions as the one used for crude RuCl_2_(azpy)_2_ catalyst. When γ-RuCl_2_(azpy)_2_ was employed, the conversion of glucose increased to 50%, and delivered **29** and **37** in 30% and 18% yield, respectively. The mediated electro-oxidation of glucose by δ-RuCl_2_(azpy)_2_ gave only 10% conversion toward **29** mainly (70% selectivity), but gluconic, glucuronic, tartaric and oxalic acid could also be observed in small quantities. These results show that γ-RuCl_2_(azpy)_2_ catalyst is the most active isomer and can deliver 2-ketogluconic acid (**37**) in a selective manner.

As previously mentioned, 2,2,6,6-tetramethyl-1-piperidinyl free radical (TEMPO) and its derivatives are very selective for alcohol oxidation. Such organic homogeneous mediators provide a metal free option of alcohol oxidation and usually shows great reactivity in mild conditions. Therefore, after having thoroughly studied the chemical oxidation of glucose with TEMPO and several co-oxidants [53,54,55]. Ibert et al., decided to investigate the TEMPO mediated electro-oxidation of glucose in alkaline medium [56]. The authors first evaluated the ability of glassy carbon and platinum to oxidise TEMPO efficiently. Platinum showed much lower reactivity toward TEMPO than glassy carbon, which also showed great stability after 10 CV experiments. Glassy carbon was therefore used as the working electrode. The chemical oxidation of carbohydrates is known to be sensitive to pH changes [53], therefore, the authors turned their attention to the effect of the pH on the electrochemical oxidation of glucose by studying the current density during CV measurements. Various carbonated solutions were used with pH ranging from 7 to 12. The clear trend was that a basic pH is beneficial to the oxidation of glucose, demonstrating the important role played by the base in such transformation. With the optimal reaction conditions in hand, it was possible for the authors to attempt a preparative TEMPO mediated electrochemical oxidation of glucose, targeting glucaric acid (**38**) as the main product. The first attempts in undivided cells showed significant limitation due to a lack of control over the parameters of the reaction. This pushed the authors to move to a jacketed reactor with a control module to regulate the pH and the temperature of the reaction mixture. In these conditions, the full conversion of glucose was achieved with 20% faradaic excess (i.e., the reaction was stopped after the total current collected exceeded the theoretical value for full conversion by 20%), however, a large amount of by-products were detected (low molecular weight products from degradation reaction) and the desired glucaric acid was only formed in poor yield. One of the main by-products detected was a tricarboxylic acid compound which had never been observed in chemical oxidation of glucose. By using GC analysis, combined with NMR studies, Ibert et al. were able to unambiguously determine the exact structure of the tricarboxylic acid (**39**) [57]. This information, combined with various analytical information, made it possible for the authors to propose a mechanism explaining the formation of **39** (Scheme 14). They postulated that the triacid resulted from a benzylic rearrangement at the C-4, C-5, and C-6 centers. This rearrangement could occur via the formation of a diketone intermediate, due to the over-oxidation of the secondary alcohols of glucaric acid (**38**). In order to prove their hypothesis, the authors performed several electrochemical oxidations with compounds such as 2-keto-gluconate, which they postulated to be intermediates in the formation of **39**.

The presence of **39** and degradation products (oxalic, tartaric, and malonic acid) in large amount, pushed the authors to reoptimize their reaction conditions. By lowering the temperature to 5 °C, and increasing the pH of the reaction to 12.2, most of the degradation and over-oxidation products formation could be inhibited (below 5% was generated). With these new optimal reaction conditions, an electrolysis of d-gluconic acid instead of d-glucose was performed, and the desired d-glucaric acid was delivered in 85% yield.

Due to its large availability and the added value of the potential reaction products, glucose has been the most studied monosaccharide for direct electrochemical transformation. Although the anodic oxidation of glucose has been studied with a range of metal (Ag, Hg, Ru, Rh, Cu, Ni, Ir, Co, Mg), most of the literature arguably focuses on the use of Au and Pt working electrodes. Indeed, both metals reactivity and selectivity for glucose electro-oxidation have first been thoroughly studied and compared via voltammetry experiments [58,59,60,61,62]. These studies gave insights on various important parameters, such as the adsorption/desorption ability of the metal electrodes [60,61,62], as well as the potential mechanisms for the direct-electro-oxidation of glucose at the surface of the electrodes [58,59]. It was found that in neutral and alkaline medium, gold electrodes displayed a greater electro-oxidation rate than any other metals, however, gold was also very prone to poisoning. To gain a deeper understanding of the electro-oxidation process and improve the performance of the electrodes, the metal surface was covered by various oxide species, or via underpotential-deposited adatoms (Pb, Tl, Bi) [27,30]. By partially covering the surface of the electrodes, Pt and Au were less sensitive to poisoning, therefore a higher current response could be observed when put in presence of glucose. However, these preliminary studies only used voltammetric measurements and did not perform long term electrolysis to test the stability and the selectivity of the electrodes for glucose oxidation. By basing their process on the results gathered by these preliminary studies, Belgsir et al. demonstrated that d-glucose could efficiently be electro-oxidised by a gold anode in alkaline medium [63]. At first, the authors employed a bare gold electrode using a three-step pulsed potential program to avoid poisoning by glucose or the reaction products. After 25 h of electrolysis, gluconic acid (**29**) could be observed in 77% yield, and glucaric acid (**38**) in trace amounts. Glucaric acid being a chemical with high added value, the authors decided to modify the surface of the gold electrode with Pb-adatoms, hoping to increase the reactivity of the electrode and thus the selectivity toward glucaric acid. The authors tested two different oxidation potential plateaus, and showed that the selectivity can clearly be changed by the addition of Pb adatoms and by tuning the potential used for the reaction (Table 4).

Although the faradaic yield decreased and more degradation products could be observed, these long term electrolysis showed that it was possible to modify the selectivity and reactivity of a gold electrode for glucose electro-oxidation by using a suitable potential program and by modifying the electrode surface with foreign metal adatoms.

Kokoh and co-workers then set out to fully investigate the influence of upd adatoms (Bi, Tl, and Pb) on both Au and Pt electrodes, in acidic, neutral, and basic mediums for the electrochemical oxidation of d-glucose [64,65]. The voltammograms recorded in acidic medium (HClO_4_, 0.1 M) showed that regardless of the electrode metal and of the adatom used, the current densities of glucose oxidation are very low, suggesting that the reactivity of glucose in acidic conditions is limited. Indeed, when the reaction mixtures were analysed after 25 h electrolysis, only a very low amount of glucose was converted to a variety of oxidation and degradation products. When voltammograms were recorded on Pt and Au anodes in neutral medium (KH_2_PO_4_/NaOH, pH = 7.3), a much greater current density was observed than under acidic conditions. However, only a limited amount of glucose was converted (up to 15%) to gluconic acid (up to 12% yield) and traces of glucaric acid with Au anode or traces of glucuronic acid with Pt electrodes. The authors then moved to investigate the behaviour of Au and Pt in presence of glucose in basic medium (NaOH 0.1 M) [64]. It is known that Au is the most efficient electrocatalyst for glucose oxidation in basic conditions, because it generates higher current densities and is less sensitive to poisoning than Pt [32,66,67]. Kokoh et al. studied the influence of the concentration of glucose (from 1 mM to 50 mM) on a pure gold anode to gain insights on the rate of the reaction and on its mechanism using voltammetry. At low initial concentration of glucose, the reaction is of order one, but at high concentration, a saturation phenomenon is observed and the overall order drops to zero. The effect of upd Tl, Pb, and Bi adatoms was also studied on a gold electrode in alkaline media. Although the voltammograms remained unchanged with and without adatoms, the current density increased during the electrolysis of glucose in presence of upd adatoms. When comparing the results of long electrolysis of glucose (**14**) performed on pure Au, Au-Bi, Au-Tl, and Au-Pb, gluconic acid (**29**) appeared to be the main product for every system. However, the Au-Pb adatom electrocatalyst also produced a significant amount of glucaric acid (**38**). This system was then studied further, by modifying the potential at which the oxidation reaction was occurring (0.5, 0.6, and 0.9 V vs. RHE). The trends observed were that the highest concentration of gluconic acid was obtained at lower potential (75% of the oxidised glucose), and that at higher potential, glucaric acid could was becoming the major product of the reaction (37% of the oxidised glucose) after 24 h of electrolysis. However, at high potential more degradation products (tartaric, oxalic, formic, and glyoxylic acid) could be observed in small quantities. The authors then looked at the glucose electro-oxidation on a Pt anode in basic conditions. After 21 h of electrolysis, 70% of glucose was converted to gluconic acid (**29**) as the main product (50% yield), and other degradation products. They then moved to electro-oxidation of glucose on a Pt electrode with Pb adatoms, using the same reaction conditions as the one used for the pure Pt electrode. A significant increase in current density was observed during the 24 h electrolysis. Within 4 h, the conversion of glucose (**14**) was nearly complete, and the concentration of gluconic acid (**29**) reached its maximum (70% yield). After the first 4 h, the gluconic acid concentration dropped to almost zero and the concentration of by-products (oxalic acid: 60%, 2-ketogluconic acid (**37**): 30%, other degradation products: traces) increased. The difference observed in the distribution of the by-products of the reaction when Pb adatoms were added to a Pt anode, lead the authors to propose possible mechanisms. For the electro-oxidation of glucose on pure Au or Pt, Kokoh and co-workers suggest that the mechanism follow similar steps (Scheme 15A). First is the adsorption/dehydrogenation of glucose at the electrode surface. Then, the free sites of the electrodes oxidise to metal-hydroxide species. The interaction of the two chemisorbed intermediates can then react and release gluconic acid.

When Pb adatoms were adsorbed on the electrode surface, d-gluconic acid (**29**) was generated rapidly and then consumed, leading to the formation of by-products, and more particularly 2-keto-gluconic acid. Taking this phenomenon into account, the authors generated the following hypothesis to explain the formation of this by-product mostly in the presence of Pb adatoms (Scheme 15B). It is postulated that the formation of gluconic acid follows the same pathway as it does with pure metal electrodes. However, once the gluconic acid is desorbed from the surface, it is possible that a new surface/chemical interaction occurs. Indeed, due to their free p-orbitals, Pb adatoms are able to coordinate with one oxygen from the carboxylic group and the oxygen from the C2 carbon. Then a free site the Pt electrode surface can dehydrogenate the C2 carbon, generating the 2-ketogluconic acid. Kokoh et al. then studied the electrochemical oxidation of d-gluconic acid at Pt and Pt-Pb adatoms anodes under basic conditions (0.1 M NaOH) [68]. After long term electrolysis with either pure Pt or Pt-Pb adatoms electrode, the distribution of products was significantly impacted by the presence of the Pb on the electrode (Scheme 16). The oxidation of gluconic acid oxidation on a pure Pt electrode appeared to be very slow and yielded mainly glucuronic acid (**40**). When Pb adatoms were adsorbed on the surface of the anode, much higher current density could be reached, and the main product of the reaction was 2-ketogluconic acid (**37**), while glucuronic acid was not observed. These observations were confirming the hypothesis that Pb was orienting the oxidation of gluconic acid on the C2 carbon by coordinating to the carboxylic acid group and the C2 hydroxyl group as shown in Scheme 15B.

Similar results were obtained when Tl adatoms were adsorbed on a Pt anode for glucose electro-oxidation [69]. Glucuronic acid (**40**) was detected in very small amount, gluconic acid was the major product of the reaction, and the concentration of 2-keto-gluconic acid (**37**) increased with the potential of the reaction (1% yield at E = −0.15 V vs. RHE; 14% yield at E = −0.02.V vs. RHE).

To deepen the understanding of the reactivity of d-glucose on a Pt electrode for anodic oxidation, Largeaud et al. set out to study the difference in behaviour of the two main dissolved anomeric forms of glucose (α and β) [70]. Indeed, the free linear aldehyde form of d-glucose is known to be the minor form in solution, and the α and β-glucopyranose forms are in an equilibrium of approximately 33% and 67% respectively. The authors evaluated the stability of α d-glucose and β d-glucose at low temperature in acidic, neutral, and basic conditions by polarimetric measurements. In acidic and neutral medium, the starting form of α or β-glucose used to make the solution remained the major anomer in solution after 5 h in solution. However, in alkaline medium, β d-glucose rapidly became the main anomer in solution, independently of the crystalline form used to make the solution. The authors then performed voltammetric measurements, in acidic and basic reaction conditions at low temperature. In basic medium, the β d-glucose being the major form in solution, it is difficult to draw conclusions on the reactivity of the α anomeric form. However, the CV measurements in acidic condition of the α and of the β d-glucose forms showed some clear difference of reactivity. Indeed, the β d-glucose showed much higher current density than the α d-glucose, suggesting a much higher reactivity of the β-anomer on a Pt anode. The hypothesis proposed by the authors is that the β conformation allows the most planar approach of the glucose molecule to the electrode surface, leading to a better interaction and a more rapid reaction. By analysing the solution during and after long term electrolysis experiments in basic medium, gluconic acid seems to be the final product of the reaction, but δ-gluconolactone could be detected and was assumed to be an intermediate during the transformation.

By using in situ reflectance infrared spectroscopic techniques (SPAIRS and SNIFTIRS), Beden et al. were able to give more details to the electro-oxidation of the β d-glucose anomer on a Pt anode in alkaline media [71]. With these investigations, the authors were able to propose a detailed mechanism for the electro-oxidation of β d-glucose in alkaline medium on a Pt anode. The first step is the chemisorption of glucose onto the surface at a free platinum site. At a potential lower than 0.6 V vs. RHE, the adsorbed dehydrated species can then be further oxidized as a weakly adsorbed gluconate via two different possible mechanisms, which depend on the potential applied to the reaction. The gluconate can easily desorb from the surface as a gluconate species (Scheme 17A). At potential higher than 0.6 V vs. RHE, the adsorbed dehydrated glucose species generates by oxidation a δ-gluconolactone species. This species can desorb and by hydrolysed to deliver gluconic acid or can remain weakly bonded to the surface while being oxidized, leading to a weakly bonded gluconate (Scheme 17B).

The ability to change the selectivity of the electro-oxidation of glucose by tuning the reaction potential was also exemplified on Au electrode by Moggia et al. [72]. Indeed, the authors studied the reactivity of d-glucose in alkaline media on a pure gold electrode, targeting d-glucaric acid as the main product of the reaction, using voltammetric measurements. In order to assign the peaks observed during CV experiments with d-glucose, they studied the response of the potential intermediates of the reaction, glucuronic acid (**40**), gluconic acid (**29**), and glucaric acid (**38**).

These measurements allowed them to clearly identify the potential at which glucose was oxidised to gluconic acid (0.55 V vs. RHE), and the potential for the oxidation of gluconic to glucaric acid (1.3 V vs. RHE). Then, by applying the appropriate potential for gluconic acid, the authors report 87% selectivity toward **29** and only trace amounts of **38** detected. When the potential was then increased to favour glucaric acid formation, the selectivity reached 14% and the selectivity toward gluconic acid decreased to 66%, thus demonstrating the relationship between the potential of the reaction and the selectivity of the outcome.

Due to the scarcity and the high price of noble metals such as Pt and Au, pure metal electrodes are not easily scalable because high costs that are involved. In order to lower the amount of metal loading on the anode, whilst preserving the selectivity and the reactivity of the metal, Holade et al. reported a selective electro-oxidation of glucose using Au nanoparticles on carbon as the anode material [73,74]. The aim of their work was to develop a highly efficient fuel cell which harvests electrical power from the oxidation of glucose, while producing added value chemicals. At first, the authors studied the reactivity of glucose toward the Au nanoparticles via CV paired with infrared analysis. These analyses revealed that glucose was successfully adsorbed on the Au nanoparticles, and could be oxidised rapidly without generating any C-C bond cleavage, and a fast hydrolysis of gluconolactone, which proved to be an inevitable intermediate. Long term electrolysis were then performed for 7.5 h under basic conditions (0.1 M NaOH). Gluconate was detected as the major product of the reaction (65% yield) with a nearly 100% faradaic efficiency. The reaction was followed by IR spectroscopy to verify that at any stage of the reaction, no C-C bond cleavage occurred.

Bimetallic nano-objects were also investigated as electro-catalyst for glucose electro-oxidation in NaOH solution (0.1 M). Rafaideen et al. recently reported the study of carbon-supported PdAu nanomaterials for selective oxidation of monosaccharides [75]. At first, the authors evaluated separately the activity of Pd and Au on carbon. They noticed that Au was significantly more active toward glucose oxidation. The authors then set out to test the electrocatalytic activity of the bimetallic Pd_x_Au_10-x_/C for monosaccharide oxidation in alkaline medium. By using voltammetric measurements, it was possible to prove that Pd rich catalysts and Au rich catalysts behaved similarly to their respective pure metal equivalents. However, the changes detected were believed to be related to the surface composition rather than to the bulk composition. It was observed that at low potential (<0.9 V vs. RHE), the activity of the catalyst increases with the increase of Au content up to 70% and then decreases, with Pd_3_Au_7_/C showing the highest catalytic activity. With the optimised reaction conditions, d-glucose (14) was electrolysed at a Pd_3_Au_7_/C anode in basic medium. After 6 h of electrolysis, 67% of glucose was consumed, and gluconate (**29**) was detected in 58% yield (selectivity towards gluconate of 87%), and very small amount of over-oxidation products could be detected. The authors then applied the same process to the electro-oxidation of d-xylose (**34**), which consumed up to 52% of xylose after 6 h of electrolysis, to deliver xylonate (**35**) in 48% yield and 92% selectivity.

Although most of the research arguably focused on the electrochemical behaviour of d-glucose, other monosaccharides could also be successfully electro-oxidised on Au or Pt electrodes. Xylose, as previously mentioned, has been electrochemically oxidised on Au and Pt electrode by Governo et al. [76]. The authors studied the catalytic activity of both metals for anodic oxidation of d-xylose (**34**) in alkaline medium using voltammetric measurements. They evaluated the effect of the concentration of xylose and of the temperature. By studying the effect of the pH, it was possible to show that on both Au and Pt, a high concentration of NaOH could contribute to the inhibition of the reaction. To avoid the electrode poisoning, a program of potentials was designed for each electrode to perform electrolysis. Au showed significantly better reactivity than Pt for the electro-oxidation of xylose (98% and 26% conversion respectively) and selectivity toward xylonic acid (**35**) (Table 5).

The oxidation of d-galactose (**30**), a C-4 epimer of d-glucose, has been reported under homogeneous and heterogeneous catalysis [77,78]. However, its electrochemical oxidation has been scarcely investigated. Parpot et al. documented the electrochemical transformation of d-galactose (**30**), and compared the selectivity and reactivity of Pt, Pt-Pb, and Au electrodes in alkaline medium [79]. For the long term electrolysis, the appropriate potential step programs were used. These potential programs were optimised with the information gathered with preliminary voltammetric measurements with the corresponding electrode. The electro-oxidation of d-galactose was carried out for 25 h at room temperature in 0.1 M NaOH solution. For the three electrodes, d-galactonic acid (**31**) was the main product of the reaction, however, Pt-Pb and Au electrode were significantly more selective toward **31** than the smooth Pt electrode, and Au was also more reactive, converting a greater amount of **30** than any other electrode (Table 6).

In this study, the authors propose a mechanism which follows the same steps as the ones described for the electrochemical oxidation of d-glucose at Pt, Pt-Pb and Au electrode (see Scheme 15A,B). The amount of galactaric acid generated during the electro-oxidation of galactose could be significantly increased by readjusting the oxidation plateau potential used [22]. Parpot et al. showed by employing voltammetry experiments that the enediol tautomer plays an important role in the oxidation of the primary alcohol of d-galatose on Au electrodes. Indeed, without this enediol form, the electro-oxidation of the primary alcohol is significantly disfavoured compared to the electro-oxidation of the aldehyde/hemiacetal group. It was also possible to show that the adsorption of the enediol species occurs at higher potential than the one used for aldehyde adsorption. Furthermore this study demonstrated that the relative position of the hydroxyl groups has an influence on the current density. Indeed, *trans*-diols are known to be more reactive than their *cis*-isomer on at Au electrode [80]. which seems to also apply for saccharides electro-oxidation. Therefore, by modifying the potential program to favour the enediol adsorption/oxidation, the authors were able to generate galactaric acid in 20% yield after 7 h of electrolysis under basic conditions (0.1 M NaOH). However, more degradation products, resulting from C-C bond cleavage, could also be observed.

The direct electrochemical oxidation of d-mannose (**32**) was also studied on Pt and Au electrodes [81]. Parpot et al. studied the behaviour of d-mannose to acidic, neutral, and basic conditions on each electrode. They found that, like d-glucose, d-mannose showed very limited reactivity toward Au or Pt in acidic media, and that the current density increased with the increase of the pH. Therefore, for the rest of the study, d-mannose was only submitted to basic conditions (0.1 M NaOH). The authors then looked at the influence of temperature and initial concentration of d-mannose. The smooth Pt electrode displayed low current density compared to Au, therefore upd Pb adatoms were also studied and proved to be very beneficial to the current density displayed. With the optimal reaction conditions in hand, d-mannose was then submitted to long term electrolysis at Au and Pt-Pb adatoms electrodes. At a Au electrode, 85% of the initial d-mannose was converted in 6 h of electrolysis and showed 49% selectivity toward d-mannonic acid (**33,** 42% yield). Beside **33**, a few low molecular weight acids arising from C-C bond cleavage could be detected in small amount (Table 7). For the direct electrolysis of d-mannose at the Pt-Pb adatoms electrode, 80% conversion of d-mannnose (**32**) was reached in 11 h. The main product was also d-mannonic acid (33, 50% yield), showing a slightly better selectivity than the pure Au electrode (Table 7).

Although Au- and Pt-based electrodes have been the most widely studied, other metal-based electrodes have been reported for the selective, direct electrochemical oxidation of monosaccharides. For instance, the electrochemical oxidation of monosaccharides at various copper-oxide-modified electrodes has been investigated, mainly for detection purposes [21,82,83]. However, some studies do report the products resulting from prolonged electrolysis [84,85,86]. The long term electrolysis of glucose on a Cu based electrode (plain Cu plate, Cu-oxides or Cu alloys) mostly resulted in the formation of one product: formate (Scheme 18). Indeed, each mol of initial glucose could be transformed via C-C bond cleavage to 6 moles of formic acid in basic media (0.1 M NaOH). Sorbitol and xylose appeared to follow the same process, generating 6 moles and 5 moles, respectively.

Certain kind of Rh porphyrins, adsorbed on a carbon electrode have also been reported for their high activity and selectivity toward glucose electro-oxidation [87]. In their study, Yamazaki et al. have employed four different porphyrin ligands to generate Rh complexes. They evaluated the performance of each catalyst by CV in alkaline medium (0.1 M NaOH) and selected the most active form of the Rh porphyrin complex, [Rh^II^(DPDE)]^+^. With the appropriate catalyst, glucose was electro-oxidised for 3 h in 0,1 M NaOH. Gluconate was the desired product in this reaction, and its concentration was determined by enzymatic method using gluconate kinase and 6-phosphogluconic dehydrogenase. Only a very small amount of gluconate could be detected (<1% yield), however a faradaic yield up to 73% was achieved due to the very low current running through the electrochemical cell.

To control the electro-oxidation of glucose to gluconic acid (**29**) and glucaric acid (**38**), Bin et al. employed an electrocatalytic reactor with a tubular porous Ti anode loaded by sol-gel method with nano-sized MnO_2_ [88]. The flow rate used during the electrolysis was adjusted to ensure that the maximum amount of d-glucose could be oxidised, while maintaining a high selectivity and avoiding over-oxidation. The effect of the loading of MnO_2_ on the Ti electrode was also investigated, as well as the pH. Surprisingly, the conversion of glucose was not strongly dependent on the pH. Indeed, from acidic (pH = 2) to basic (pH = 10), the conversion was always above 90%. However, the selectivity toward **29** and **38** appeared to be improved when the pH was neutral. The temperature also had an impact on the selectivity but also on the overall conversion of glucose, which significantly decreased when the reaction was performed at a temperature above 30 °C. The authors explain this observation by the fact that the adsorption process is unfavoured with high temperatures. Performing the electrolysis at the most favourable conditions, 93% of the initial d-glucose was converted mainly to both gluconic acid (42% yield) and glucaric acid (44% yield).

Nickel-based electrodes have also been investigated for the electro-oxidation of mono-saccharides. Indeed, Parpot et al. first documented the direct electrochemical oxidation of d-galactose (**30**) on a nickel electrode in alkaline medium (0.1 M NaOH) [79]. Two types of long term electrolysis were carried out: one with a constant potential and one with a step potential program. In both cases, the conversion of d-galactose was low (17% and 43% respectively). Although d-galactonic acid (**31**) was detected as the major product of the reaction, only small amounts were generated (9% at constant potential and 7% with the plateaus potential program). These electrolysis also produced various low molecular weight acids (formic, glycolic, oxalic acid). The poor yield of **31** can be explained by the competition reaction between the desired oxidation, the degradation responsible for low molecular weight compounds formation, and water oxidation. The oxidation of galactose on a Ni anode occurs near the oxygen evolution potential region. This suggests that the surface is covered with NiOH and NiO species which can react with galactose (or galactonic acid) and generate the C-C bond cleavage products (Scheme 19).

Similar results were observed for the electrochemical oxidation of d-mannose (**32**) on a Ni electrode under basic conditions (0.1 M NaOH) at constant potential [81]. The conversion of d-mannose was very low (35%), and a significant amount of degradation products were presents, some in larger quantities than the desired d-mannonic acid (**33**, 3% yield).

More recently, Liu et al. reported that nanostructured NiFe oxides (NiFeO_x_) and NiFe nitrides (NiFeN_x_) catalysts exhibit a great activity and selectivity toward the electrochemical oxidation of d-glucose (**14**) [89]. The authors used nickel foam as a support as a Ni source for the catalyst and to ensure a 3D structure to the electrode. Then, the resulting oxides and nitride electrodes were tested in 1.0 M KOH solution (pH = 13.9), for d-glucose oxidation. The preliminary tests showed that the electro-oxidation of glucose happened at a lower potential than the oxygen evolution reaction, which was presumed to be the main undesired side reaction. NiFeO_x_ appeared to be more active than NiFeN_x_ toward glucose oxidation. By analysing the electrode materials after electrolysis of glucose, NiOOH and FeOOH species were detected. The authors postulated that these species are the catalytic active sites of the electrodes responsible for the glucose electro-oxidation reaction. Then, they turned their attention to evaluating various parameters such as the reaction potential and the initial glucose concentration. The authors were able to carry out long term electrolysis with both catalysts. Both NiFeO_x_ and NiFeN_x_ converted glucose very efficiently (91% and 93% respectively), and with great selectivity toward glucaric acid (**38**, 71% and 64% yield respectively) after 18 h. The monitoring of the reaction allowed the authors to see the initial formation of gluconic acid (**29**) and then its conversion to glucaric acid (**38**). The catalysts also showed great stability, when five chronoamperometric measurements were carried out the conversion of glucose slightly decreased but the faradaic efficiency remained almost unchanged. By using IR and NMR spectrometry, the authors were able to identify various intermediates and products of the reaction, and proposed the following reaction pathway for the NiFe anodic oxidation of d-glucose (Scheme 20).

### 2.4. Summary

The electrochemical oxidation of saccharides has attracted a lot of attention in the past century. Although it is difficult to compare different studies between themselves because they all use different reaction parameters, it seems necessary to clarify which electro-catalysts seems promising and for which time of transformation. The time of reaction being different in each example is not taken into consideration, as well as the pH of the reaction. However, it is worth noting that the electrochemical oxidations of saccharides are almost exclusively performed at high pH. Indeed, at low pH, the saccharide are likely to degrade through cleavage of the C-O-C bond or the opening of the ring. The few electro-oxidations documented to happen in acidic medium mainly lead to the corresponding hydration products (for poly- and di-saccharides) as well as degradation products.

For the indirect electro-chemical oxidation of saccharides, the most explored and promising mediator is TEMPO. It has been successfully used on poly-, di-, and monosaccharides. TEMPO selectively oxidises primary alcohols and is able to deliver triacids when used on disaccharides such as d-raffinose with 63% yield, but also with cellobiose and d-maltose, although less selectively. Dicarboxylic acid could also be obtained in 61% when TEMPO was used to oxidise trehalose. The TEMPO mediated electrochemical oxidation of monosaccharides such as d-glucose yielded the corresponding mono- and disaccharide products and the selectivity could be tuned to favor one or the other.

A vast number of metal electrodes has been investigated for the direct electro-oxidation of saccharides, and it is therefore complex to identify which is the most promising, the most reactive, or the most selective. However, it is very clear that Au is the most investigated metal. It was tested with poly-, di-, and monosaccharides, with and without adatoms. Pure Au plates were used as well as nanoparticles. In all reports, the major product isolated with Au as the electrocatalyst is the mono carboxylic acid equivalent of the starting saccharide. Pt was also extensively studied and demonstrated similar selectivity as gold, but usually lower reactivity. When a NiFe electrocatalyst was used on d-glucose, the major product isolated was d-glucaric acid (71% yield), therefore showing a different selectivity than the other catalysts. Finally, when Cu was used as electrocatalyst with monosaccharides, formate appeared to be the only product formed quantitatively (Table 8).

## 3. Electrochemical Reduction of Biomass

### 3.1. From Polysaccharides to Monosaccharides

The electrochemical reduction of saccharides has been studied extensively since the early 1900s., but arguably, most of the effort in this domain has been focusing towards the electroreduction of d-glucose (**14**) toward the corresponding alditol: sorbitol (**41**). Alditols have found a wide range of applications across various industries [90]. nevertheless, their current way of production is via catalytic hydrogenation processes which require high temperature and high pressures [91]. Therefore, the development of a greener, more user friendly, and sustainable way to produce such polyols is greatly sought after.

Amongst the first examples of electrochemical reduction of glucose, it was found that under acidic reaction conditions, a lead cathode does not generate the desired alditol but d-arabinose and formaldehyde [38]. Other studies focused on monosaccharides electroreduction under basic or neutral reaction conditions. Creighton et al. reported the successful electrochemical transformation of d-glucose (**14**), d-mannose (**32**), d-fructose (**13**), and d-xylose (**34**) to the corresponding alditols in neutral medium (Na_2_SO_4_) [92]. Preliminary tests were performed on the electroreduction of d-mannose (**32**) on a Hg cathode. The authors evaluated various parameters such as the temperature, the initial concentration and the pH. Mannitol (**42**) could be isolated in 62% yield. Despite these promising results, the use of an amalgamated Pb cathode instead of Hg was explored, due to the numerous disadvantages Hg brings. d-glucose was then used for the preliminary tests. Due to isomerisation equilibrium between glucose, fructose and mannose, the authors anticipated that it could be possible to generate not only sorbitol from glucose, but also mannitol (Scheme 21). After screening the same parameters as previously used, it was clear that the hydroxyl ion concentration in the catholyte influenced greatly the selectivity of the reaction. Indeed, at low concentration of NaOH (2.5 mM), no mannitol was generated and sorbitol was isolated in 59% yield. However, with a high concentration of NaOH (0.75 M), **42** was isolated in 13% yield, and **41** was isolated in 47% yield. The authors also reported the successful conversion of xylose (**34**) to xylitol, galactose (**30**) to dulcitol, and lactose (**25**) to a mixture of dulcitol and sorbitol, however, no information was given on the selectivity and the efficiency of these electrochemical reductions. From the range of parameters tested and by varying the reduced saccharides, the authors postulated that Pb and Hg were good cathode materials for poorly reducible compounds due to their high overpotential for hydrogen evolution.

The electrochemical reduction of saccharides was used industrially to generate the corresponding alditols. However, a range of undesired by-products were also present, depending on the reaction conditions. Wolfrom et al. documented a very thorough study to characterise the various products present after the electro-reduction of glucose on a Hg cathode [93,94,95,96,97]. From the commercial sorbitol (**41**) in weakly alkaline medium (pH = 7–10) and at a temperature below 30 °C, d-mannitol (1% yield), and 2-deoxysorbitol (**43**, 5% yield) could be isolated. When glucose was electro-reduced to sorbitol in highly basic conditions, (pH = 10–13), a more complex mixture was present at the end of the reaction. As expected, sorbitol (**41**) and mannitol (**42**) were isolated, as well as 2-deoxysorbitol (**43**) and 1-deoxymannitol (**44**). d,l-glucitol [96] and a dodecitol [97] were also detected. The formation of dodecitol was assumed to arise from the electroreduction of the aldehyde group of two glucose molecules.

The use of alkaline electrolytes is beneficial for the reactivity of the electrodes for the reduction of monosaccharides but is detrimental to the selectivity of the reaction because it favours the isomerisation reaction of glucose to mannose and fructose. By using sulphate solutions as electrolyte, this problem was solved. Despite their nearly neutral pH, the hydrogen evolution reaction at the electrode surface generates a layer of higher alkalinity, which is favourable for the electroreduction of glucose to sorbitol. Rice and co-workers reported that, in such condition and on a Pb cathode, sorbitol was the major product detected after only 4 h of electrolysis, and the concentration of the undesired fructose and maltose started increasing only after longer electrolysis [98]. The authors reported in their study the influence of various parameters such as the pH, the initial concentration of glucose, and the flow rate on the current density at 20 °C. The addition of Zn^2+^ in the catholyte had been used industrially to improve the rate of the reaction. The authors investigated how the Zn^2+^ ions impacted the reaction. They observed that the Zn was deposited on the electrode surface, which increased the overall surface area of the electrode, thus the rate of glucose electroreduction will increase. The authors also hypothesized that Zn could have a dual effect by acting as a co-catalyst for the electroreduction reaction.

The electrochemical reduction of glucose on Zn on carbon nanotubes (CNTs) grown on a graphite electrode was documented by Fei et al. [99]. At first, the authors studied the variation of reactivity between a flat Zn, a Zn/graphite, and a Zn/CNTs electrodes in weakly alkaline medium (0.1 M Na_2_SO_4_, pH = 11) using CV measurements. The flat Zn and the Zn/graphite electrodes displayed similar behaviours, but the Zn/CNTs electrode showed an increase of the reduction current in the presence of glucose, due to the high surface area provided by the CNTs. The 3D structure of the CNTs provided a much larger zone of higher alkalinity in the solution than on a flat electrode, which is known to be beneficial for the conversion of the cyclic hemiacetal form of glucose to its electroreducible aldehyde form, thus explaining the higher efficiency of the Zn/CNTs electrode to reduce glucose. However, the electrode stability was very low (50% loss of efficiency in 4 h), probably due to the dissolution of Zn on CNTs which happens during the electrolysis of glucose. In order to improve the electrode stability, Zn-Ni and Zn-Fe alloys were deposited on CNTs. Both electrodes display similar reactivity as the Zn/CNTs electrode, but with slightly higher hydrogen evolution current. The Zn-Fe/CNTs and Zn-Ni/CNTs electrodes showed better stability than the Zn/CNTs electrode. Indeed, after 8 h of electrolysis, the current decreased but significantly less than for the Zn/CNTs electrode, proving that the activity of the electrode was maintained. The authors noticed that the current efficiency of the Zn-Fe/CNTs electrode was higher for glucose electroreduction than all the other electrodes, thus they investigated further the reaction with the Zn-Fe/CNTs electrode only. They studied the effect of the pH on the electrode response toward glucose, as well as the temperature. Although an increase of pH (from 7 to 13) shows a drastic increase in current, the authors acknowledge that the isomerisation of glucose to fructose and mannose will have a negative impact on the current efficiency. They report the highest current efficiency at 40 °C. Their study was thus able to prove the benefices of the utilisation of Zn in combination with CNTs for the electro-transformation of glucose to sorbitol.

The most commonly used metals for the electrochemical reduction of glucose are the metals which suppress the hydrogen evolution reaction (HER), where the chemisorbed hydrogen on the surface of the electrode is produced by the electro-reduction of water and then produce hydrogen gas, which is in direct competition with the hydrogenation of the adsorbed glucose to yield sorbitol or deoxygenate to deoxysorbitol. Kwon et al. documented a very thorough study where a vast number of pure metal electrode catalysts were screened in order to comprehend the activity and the selectivity of catalysts for the electro-reduction of glucose [100]. All the tests were performed in neutral electrolyte, and the metals were separated in three groups, metals that generate sorbitol (**41**) and hydrogen gas (Fe, Co, Ni, Cu, Pd, Ag, Au, and Al); metals forming H_2_, sorbitol (**41**) and 2-deoxysorbitol (**43**) (Zn, Cd, In, Sn, Sb, Pb, and Bi); and metals forming only hydrogen gas (Ti, V, Cr, Mn, Zr, Nb, Mo, Hf, Ta, W, Re, Ru, Rh, Ir, and Pt). The metals were tested via voltammetric measurements, while the products in solution were analysed via an online HPLC analysis (Table 9).

First, the reactivity and selectivity of Fe, Co, Ni, Cu, Pd, Ag, Au, and Al electrodes were investigated in 0.1 M Na_2_SO_4_ (entries 1 to 8). All these metals did convert glucose to sorbitol exclusively, but also performed the hydrogen evolution reaction. Most of these metals showed a lower reduction current when put in presence of glucose. This observation indicates that the adsorption of glucose on the surface of the electrode blocks the hydrogen evolution reaction by competing for the same active sites on the electrodes. Au and Cu electrodes were the two exceptions and displayed a higher reductive current in presence of glucose. These two metal electrodes also appeared to be the most efficient at converting glucose to sorbitol. For the formation of sorbitol, Ni is the metal which has the lowest onset potential compared to all the other metals (entry 3), and Cu gave the highest conversion to **41** of this series of metals (entry 4).

The second series of metals investigated (entries 9 to 15) were the metals able to generate sorbitol (**41**), as well as 2-deoxysorbitol (**43**). Higher cathodic currents could be applied to these metals, and therefore a higher conversion of glucose could be reached. The cathodic current generated by Zn was higher than the one generated by cadmium. Cd proved to be much more effective at converting glucose to **41** and **43** (entries 9 and 10, respectively). This observation suggests that Cd is more effective than Zn to hydrogenate glucose by suppressing the hydrogen evolution reaction and has a higher selectivity toward sorbitol. Sb showed the lowest onset potential (entry 13), Pb the best yield (entry 14), and Sn gave the best selectivity toward sorbitol (entry 12).

The final series comprise no active metals for the electrochemical reduction of glucose and are therefore not discussed here.

Acidic reaction condition (0.5 M H_2_SO_4_) proved to be inadequate for glucose hydrogenation despite the use of metals such as Co, Pb, or Cd, which were active in neutral medium. Therefore, the presence of OH- species is necessary for the electrochemical hydrogenation of glucose. However, having an alkaline electrolyte is detrimental to the selectivity of the reaction by promoting the isomerisation of glucose to fructose and mannose, and therefore to various side products, proving that using an unbuffered neutral electrolyte is the best reaction conditions for efficient and selective electrocatalytic glucose reduction.

When high cathodic currents were used, fructose (**13**) could also be detected in the solutions, which emphasizes the important of local pH at the electrode surface. Indeed, the rate of formation of OH^−^ species is controlled by the current intensity. With the information gathered during the investigation on the metal electrode screening, the authors were able to propose a mechanism explaining the formation of the various products and by-products (Scheme 22). The authors based their assumption on previous work which described that the cleavage of C-OH bond was favoured during the deoxygenation reaction when the hydroxyl group was vicinal to a C=O bond [101]. From this fact, glucose (**14**) itself could be a direct precursor to 2-deoxysorbitol (**43**), and would not require the formation of fructose (**13**) as an intermediate as it was previously postulated. Furthermore, the absence of mannitol (**42**) supports the scenario supporting the direct deoxygenation of glucose (**14**) to generate 2-deoxysorbitol (**43**). Indeed, the prochiral C2 position in fructose (the ketone) can be hydrogenated to two isomeric hydroxyl groups in almost equivalent ratios 40% sorbitol, 60% mannitol) [102].

Although most of the work on saccharide electro-reduction has been directed to the reduction of d-glucose, other monosaccharides have also been studied. d-xylose was successfully converted into its corresponding alditol, xylitol [103]. Piszczek et al. first optimised the reaction conditions by variating the pH and the temperature of the reaction. They reported that by electrolysing xylose (**34**) on an amalgamated Pb cathode in a alkaline solution (Na_2_SO_4_, pH = 11) and at 45 ᵒC, the desired xylitol (**44**) could be isolated in 75% yield. Small amounts of 2-deoxyxylitol could also be detected (Scheme 23).

Hamann et al. documented the formation of ester derivatives of sucrose and various protected monosaccharides [104,105]. The reaction was a two-step process, first the saccharides were electrochemically reduced to their corresponding anion, and then were reacted with the alkyl halide to deliver the desired saccharide ester derivative (Table 10).

The reaction took place in DMF in presence of LiBr. The Li ion was necessary to stabilise the anionic saccharide species generated during the electrolysis. The authors screened other metal counter cations, (Mg and Zn), however, the deprotonation did not take place. When sucrose was submitted to such reaction conditions, a poor selectivity was observed. Indeed, three hydroxyl groups reacted and therefore three different products could be detected. Nevertheless, the substitution at the C2 position appeared to be the most favoured one.

### 3.2. Summary

The electrochemical reduction of saccharides has arguable been less documented than the electro-oxidation process. However, the various reports demonstrate the feasibility and scalability of such processes. Furthermore, a range of product selectivity can be obtained by modifying the reaction parameters such as the electrode material, the supporting electrolyte, and the pH (Table 11).

## 4. Combined Electrolysis

During electrolysis, oxidation and reduction reactions are happening simultaneously on both the anode and cathode respectively. When oxidation and reduction reactions are coordinated in such way that they both yield targeted organic compounds, this is often referred to as ‘paired electrolysis’. This can be performed in an undivided reactor by using the same starting material which can be reduced and oxidised and deliver two different products, or in a divided reactor, with one or two different starting materials, thus delivering one oxidised and one reduced product (Scheme 24). This type of electrochemical reactions are intensely pursued due to the high energy efficiency relative to the amount of chemicals generated.

Park et al. documented a paired electrolysis of d-glucose to gluconate (**29**) at the anode and simultaneously to sorbitol (**41**) at the cathode in an undivided flow reactor [106,107]. Graphite chip was used as the anode material for an indirect electro-oxidation and either a Zn(Hg) shot or Raney Ni as the cathode for a direct electro-reduction reaction. The anodic oxidation of glucose to gluconate was mediated by NaBr as previously described by Isbell and co-workers [40,41,42]. To push the reaction toward the consumption of the intermediate δ-gluconolactone, the paired electrosynthesis was carried out at 58 °C with a residence time of 10 min in the reactor and at pH 7. The use of a Raney-Ni electrode allowed to reduce the overall cell potential compared to the Zn(Hg) electrode. Furthermore, when Raney-Ni and graphite were associated as the cathode and the anode material respectively, the current efficiency for both transformations could reach 100%, but the glucose conversion remained low (<30%). The authors also observed that Raney-Ni get poisoned during the electro-reduction of glucose due to the hydrogen evolution reaction. An in situ catalyst regeneration was then possible by washing the electrode with a 17 wt% NaOH solution at 60 °C for 90 min [107]. This procedure allowed to restore the catalytic activity of the cathode and to restore the current efficiency for sorbitol from 35–45% to 70–100%.

In a similar process, Bardot and co-workers developed a paired electrolysis of fructose (**13**) to gluconic acid (**29**) at the anode and to sorbitol (**41**) and mannitol (**42**) at the cathode in a membrane separated reactor [102]. In this case, the anode was a graphite electrode and the cathode was a graphite electrode coated with Pt and Rh metals. This study only focused on the electrochemical reduction of fructose to sorbitol and mannitol, and the authors report that fructose was converted up to 60% and was transformed exclusively to sorbitol (40% selectivity, 24% yield) and mannitol (60% selectivity, 36% yield).

Finally, d-xylose (**34**) was successfully anodically oxidised to xylonic acid (**35**) and reduced to xylitol (**44**) at the cathode in a membrane separated reactor [108]. First, the authors investigated the cathodic reduction of xylose on an amalgamated Zn electrode. However, for the voltammetric measurements performed, they could deduct that the reduction of xylose was heterogeneously catalysed by adsorbed H-adatoms on the surface of the electrode. To promote this catalytic process, a variety of metal coating were screened (WFe_3-x_Pt_x_ and MoFe_3-x_Pt_x_, and other coating with Co or Ni instead of Fe or with Ru instead of Pt). With such coating, the H-adatoms catalytic reduction of xylose was significantly faster, the cathodic potential was reduced compared to the Zn(Hg) electrode and the Faradaic yield for xylitol formation was exceeding 90%. For the anode reaction, RuO_2_/TiO_2_ and Pb/Sn were used as intermetallic coating on a Ti electrode. These modified electrodes were used in combination with bromine as mediator for the electro-oxidation of xylose. This process displayed high anodic current efficiency (>97%) and the Pb/Sn coating shifted the potential for oxygen evolution to higher potential, improving the selectivity toward the anodic oxidation of xylose. Bromide species which are essential to the anodic formation of **35** and detrimental to the cathodic reaction reduced the cathodic Faradaic yield below 60%. Therefore, in order to maintain the highest efficiency possible of both side of the reaction, the authors emphasised on the necessity to separate this process with a membrane.

## 5. Conclusions

Research on electrosynthesis in general has come a long way, with some applications more advanced than others and a huge acceleration due to the emerging electrification of the chemical industry [109]. Electrosynthesis of biomass-based feedstocks is not very straightforward and depends on several process parameters such as pH, temperature, concentration, and potential as well as on the choice and availability of components like stable electrocatalysts and membranes. In this review, we have summarized the recent state of the art for oxidation and reduction of mono-, oligo- and polysaccharides. The main electrocatalyst used in these conversions were summarized and a basic screening reveals that most often used ones are based on pure metals like gold and platinum. This trend is changing in recent years when more metals, carbons, alloys, oxides and bimetallic catalysts are being explored. The need of the hour is moving from brute screening to smart designing and screening of the electrocatalysts coupling density functional theory (DFT)-based modelling with high throughput screening. Currently this is still based on a trial and error approach which is costly and time consuming due to a myriad of possible catalysts and shapes thereof. Besides, we noticed that most reported studies focus on one or a few ‘promising’ metals. Although electrolysis is often hyped as providing a ‘highly selective’ reaction pathway, the review shows that this statement is flawed in many cases. Even when starting from pure substances (i.e., disregarding the complexities of real-life biobased feedstock), formation of side-products is oftentimes difficult to avoid, resulting in poor selectivities and ultimately a process that is challenging to become economically viable.

Standardisation in terms of reporting the results is an important issue to be tackled and should address which are the most important performance numbers for evaluation of the electrosynthesis process. More standardization is needed in terms of reporting on yields, efficiencies, selectivity and other process parameters, the absence of which makes the comparison between different studies difficult. Even among two similar studies, the reported results are not always consistent. The research on this topic is rather fragmented research with many fundamental knowledge gaps (such as missing reaction mechanisms). Even though electrosynthesis has shown promise, it still needs to conquer a lot of ground in terms of performance and scalability.

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
