# Peer review of "Electrosynthesis of Biobased Chemicals Using Carbohydrates as a Feedstock"

_molecules, 2020, doi:10.3390/molecules25163712_

Round 1

Reviewer 1 Report

This is a good review of the literature but requires some modifications to make it more readable and useful. As is, the manuscript is too dense and difficult to extract information without a lot of effort.

Abstract: Can the authors include key take-aways from the review in the abstract?

The authors have done a good job of compiling relevant literature on the conversion of simple carbohydrates to various products. The information is rich in chemistry, but lacks identification of promising candidates for moving them further along. The conclusion discusses a few ideas but it would be a good idea to expand that in a pre-conclusion section. The review is lengthy and extraction of information useful to readers in a summarized fashion will be helpful.

The authors have given a number of reactions and schemes, but it is unclear how the reaction mechanisms are influencing the outcomes. Are there specific mechanisms in action for particular redox reactions? They have referred to some mechanisms which are scattered throughout the review, but they are not extracted out with any common principles. If possible, this should be attempted.

Another criteria which can result in analytical data mining is categorization of key catalysts for groups of products that can be produced via electro-conversion, such as diacids, triacids, mono/di/trialcohols, aldehydes, etc.   

Below are some minor observations:

Line 292: There seems to be a typo in the word: accruing?

Line 314: Typo in sentence: ‘…converted in 70% exclusively…’

Line 925: Typo in word : bloc

Author Response

Reviewer: This is a good review of the literature but requires some modifications to make it more readable and useful. As is, the manuscript is too dense and difficult to extract information without a lot of effort.

 Authors: Thank you for this comment. We have included some pre-conclusion sections after each major section in order to make it easier to extract the desired information.

Abstract: Can the authors include key take-aways from the review in the abstract?.

Authors: The abstract was modified with brief sentences summarising the general conclusions drawn from the review (line 17 to 21).

Reviewer: The authors have done a good job of compiling relevant literature on the conversion of simple carbohydrates to various products. The information is rich in chemistry, but lacks identification of promising candidates for moving them further along. The conclusion discusses a few ideas but it would be a good idea to expand that in a pre-conclusion section. The review is lengthy and extraction of information useful to readers in a summarized fashion will be helpful.

 Authors: Explanatory summaries, as well as table 8 and table 11 have been added to simplify the visualization and the comparaison of the various processes.

Reviwer: The authors have given a number of reactions and schemes, but it is unclear how the reaction mechanisms are influencing the outcomes. Are there specific mechanisms in action for particular redox reactions? They have referred to some mechanisms which are scattered throughout the review, but they are not extracted out with any common principles. If possible, this should be attempted.

Authors: We agree that having a general mechanism which could explain the outcomes of the redox reaction would have been of great interest, however, the variables are too numerous. It is therefore impossible to summarise into one general mechanism.

Reviewer: Another criteria which can result in analytical data mining is categorization of key catalysts for groups of products that can be produced via electro-conversion, such as diacids, triacids, mono/di/trialcohols, aldehydes, etc.

Authors: Thanks a lot for this suggestion. We have now included a summary table (Table 8 and table 11 in the revised manuscript) to illustrate this point.

Reviewer: Below are some minor observations:

Line 292: There seems to be a typo in the word: accruing?

Line 314: Typo in sentence: ‘…converted in 70% exclusively…’

Line 925: Typo in word : bloc.

Authors: We have corrected all the typographical mistakes in the revised manuscript as suggested by the reviewer. Also we have ensured that the SI units have been used everywhere.

Reviewer 2 Report

Dear Authors,

The manuscript entitled “Electrosynthesis of biobased chemicals using carbohydrates as a feedstock” requires several modifications, before it can be recommended for consideration for publication, as follows:

Fig 1 should be changed to Scheme 1, as it is Scheme and not a figure. Also the colours should be improved because in the present form it is not clear and doesn’t seem to be visually interesting.

In Scheme 2 what means the “±” in the “± glassy carbon”? I suggest to remove it as it is confusing. Same in Scheme 11, Table 9 and so on.

What is the horizontal dotted bar referring to in Scheme 3A? I would suggest to change the bars into schematic that presents a pot or something adequate.

Why the synthesis is incorporated into the Table 1. I suggest to separate those two. Same in Table 3, 5 and so on.

In Scheme 8, the 0.53V vs. SCE should be removed.

In Scheme 9, the scheme is drawn as a cycle leading finally to Au, which I find questionable. Can Authors explain why is it presented in such a way? Otherwise I suggest this cycle to end on the product 26.

In Scheme 10. The comments (90% conversion and 95% selectivity) should be deleted. Only synthesis parameters should be given.

Why some of the Tables show compounds listed by name, other name and structural pattern and another using numbers. I would suggest to unify the form of the data presentation within the manuscript.

Equation 1, I would suggest to delete it as it is not very informative and looks trivial. The equation can be summarized in the main text.

In Table 4, the heading text should not be cut therefore I suggest to replace the columns with rows, so that the text can be readable.

In Scheme 14 delete the dotted line between A) and B). It is clear which s A and which is B.

I do not understand Scheme 17 graphical representation.

Author Response

Reviewer: The manuscript entitled “Electrosynthesis of biobased chemicals using carbohydrates as a feedstock” requires several modifications, before it can be recommended for consideration for publication, as follows:

Fig 1 should be changed to Scheme 1, as it is Scheme and not a figure. Also the colours should be improved because in the present form it is not clear and doesn’t seem to be visually interesting.

Authors: We agree and we have renamed the Fig 1 to Scheme 1. Accordingly the other figure numbers have also been corrected to reflect this change. We also agree that the color scheme in the original submission was not clear. We have revised this Scheme to make it more clear and legible. The revised figure now looks like this-

 SEE ATTACHED FILE

Reviewer: In Scheme 2 what means the “±” in the “± glassy carbon”? I suggest to remove it as it is confusing. Same in Scheme 11, Table 9 and so on.

Authors: The sign “±” has been deleted from every scheme where it was present (Scheme 3, ligne 115; Scheme 7, ligne 225; Table 3, ligne 451; Scheme 12, ligne 462; Table 10, lign 1067

Reviewer: What is the horizontal dotted bar referring to in Scheme 3A? I would suggest to change the bars into schematic that presents a pot or something adequate.

Authors: The dotted bar was originally present to calrify the separation between scheme 3A and 3B; however, due to formatting, the dotted bar moved and was not usefull anymore. The dotted bar was therefore deleted (ligne 139)

Reviewer: Why the synthesis is incorporated into the Table 1. I suggest to separate those two. Same in Table 3, 5 and so on.

Authors:  The synthesis is included into some table in order to exemplify which reaction is described. The synthesis is associated with a table when no other schemes are present to help at the visualization of the reaction.

Reviewer: In Scheme 8, the 0.53V vs. SCE should be removed.

Authors: This has been corrected as suggested and this text has been removed.

Reviewer: In Scheme 9, the scheme is drawn as a cycle leading finally to Au, which I find questionable. Can Authors explain why is it presented in such a way? Otherwise I suggest this cycle to end on the product 26.

Authors: The scheme 10 (previously scheme 9) line 388 is representing the catalytic cycle responsible for the transformation of D-lactose, it therefore follows the various oxidation states of the catalyst along the transformation. It is our believe, based on the report of the authors of the original study, that the Au hydroxide involved in the oxidation of lactose regenerate Au0 after the reductive elimination step necessary for the formation of the lactobionic acid. Therefore, the catalytic cycles have to lead back to the starting state of Au.

Reviewer: In Scheme 10. The comments (90% conversion and 95% selectivity) should be deleted. Only synthesis parameters should be given.

Authors: Corrected as suggested in the revised manuscript and the text within the parantheses is deleted (Line 410).

Reviewer: Why some of the Tables show compounds listed by name, other name and structural pattern and another using numbers. I would suggest to unify the form of the data presentation within the manuscript. 

 Authors: The tables showing the chemical structures are describing the scope of a reaction process, it is thus important to visualize which are the major prodcuts generated by such methodology. The tables showing only a list of compound names are included to demonstrate the level of selectivity of the reaction described, the list of compounds can be sometimes quite extensive and the chemical  structures of the by-products fromed were not necessary to exemplify the lack or the good selectivity of the process.

Reviewer: Equation 1, I would suggest to delete it as it is not very informative and looks trivial. The equation can be summarized in the main text. 

Authors: Equation 1 has been deleted from the revised manuscript (line 473)

Reviewer: In Table 4, the heading text should not be cut therefore I suggest to replace the columns with rows, so that the text can be readable. 

Authors: We agree with this suggestion and have revised the Table  (line 575)

Reviewer: In Scheme 14 delete the dotted line between A) and B). It is clear which is A and which is B.

Authors: We discussed it and politely disagree. We believe this clear demarcation makes the scheme neat and clearly separats part A and B.

Reviewer: I do not understand Scheme 17 graphical representation. 

Authors: The scheme 17 (Scheme 18, line 802 after revision) has been modified in order to clarify the process described. It represents the electrochemical formation of formate form glucose. Each arrow represents a contact between the the substrates and the Cu electrode. After each step, a carbon atom is removed from the original substrate until the formation of the 6th and last molecule of formate.

Reviewer 3 Report

This paper reviews the current studies on electrosynthesis of biochemicals from carbohydrate. It is an interest topic and would attract many readers. However, the current quality of this manuscript is not suitable for publication. It seems like a summary of each study and put together without interpretations. All updated studies should be organized to connect each other and discussed all together in a point of view. Awkward paragraph separations were found in many parts of the manuscript.

Author Response

Reviewer: This paper reviews the current studies on electrosynthesis of biochemicals from carbohydrate. It is an interest topic and would attract many readers. However, the current quality of this manuscript is not suitable for publication. It seems like a summary of each study and put together without interpretations. All updated studies should be organized to connect each other and discussed all together in a point of view. Awkward paragraph separations were found in many parts of the manuscript.

Authors: We thank the reviewer for his feedback and happy to hear that he/she found the topic interesting. As for the other point is concerned, we have now thoroughly revised the original submission based on the feedback received from the Editor and the 4 reviewers. The studies are now well connected to make a cohesive story and some of the sections have been summarized better. A pre-conclusion section has been added which gives in detail the main outcomes of the studies. Two new tables (TABLE 8 and 11) has been added to connect all the catalysts used with the primary substrate and the final products. Some of the complex reactions have also been described in greater detail.

Reviewer 4 Report

Electrosynthesis of biobased chemicals using carbohydrates as a feedstock, Vincent Vedovato, Karolien Vanbroekhoven, Deepak Pant and Joost Helsen.

Comment 1. The abstract is clear, interesting, and well explained.

Comment 2. Line 28. “This is especially true of Europe where bio-based industries are central to building a European circular economy” should be replaced by “This is especially true in Europe where bio-based industries are central to build a European circular economy”.

Comment 3. Line 37. “Even though carbohydrate chemistry has been a subject of study since the 19th century following from Fisher’s work but has arguably received limited attention compared to other biomolecules such as proteins or amino-acids” should be replaced by “Even though carbohydrate chemistry has been a subject of study since the 19th century following from Fisher’s work, it has arguably received limited attention compared to other biomolecules such as proteins or amino-acids”.

Comment 4. Line 49. “due to decoupling the chemical reaction in half reactions” should be replaced by “due to the decoupling of the chemical reaction in half-reactions”.

General comments

The English is in need of minor improvement, and some language changes are included in this revision only as examples.

I find the manuscript relevant, complete, detailed, and very interesting.

Author Response

Reviewer: The abstract is clear, interesting, and well explained..

Authors: Thank you. We have tried our best to improve the manuscript further.

Reviewer: Line 28. “This is especially true of Europe where bio-based industries are central to building a European circular economy” should be replaced by “This is especially true in Europe where bio-based industries are central to build a European circular economy”.

Authors: The suggested sentence has been revised as suggested.

Reviewer: Line 37. “Even though carbohydrate chemistry has been a subject of study since the 19th century following from Fisher’s work but has arguably received limited attention compared to other biomolecules such as proteins or amino-acids” should be replaced by “Even though carbohydrate chemistry has been a subject of study since the 19th century following from Fisher’s work, it has arguably received limited attention compared to other biomolecules such as proteins or amino-acids”..

Authors: We agree and have revised the sentence accordingly.

Reviewer: Line 49. “due to decoupling the chemical reaction in half reactions” should be replaced by “due to the decoupling of the chemical reaction in half-reactions”.

Authors: Thank you for this comment. We have now revised this sentence.

Reviewer: The English is in need of minor improvement, and some language changes are included in this revision only as examples.

Authors:  Thank you. We have tried our best to improve the language along with the remaining typos.

Reviewer: I find the manuscript relevant, complete, detailed, and very interesting.

Authors:  Thank you. We are motivated by such feedback.

Round 2

Reviewer 3 Report

The manuscript was significantly revised and can be accepted for publication.